# *GL5.2*, a Quantitative Trait Locus for Rice Grain Shape, Encodes a RING-Type E3 Ubiquitin Ligase

**DOI:** 10.3390/plants13172521

**Published:** 2024-09-08

**Authors:** Hui Zhang, De-Run Huang, Yi Shen, Xiao-Jun Niu, Ye-Yang Fan, Zhen-Hua Zhang, Jie-Yun Zhuang, Yu-Jun Zhu

**Affiliations:** 1Crop Research Institute, Fujian Academy of Agricultural Sciences, Fuzhou 350013, China; zhanghui-zws@faas.cn; 2State Key Laboratory of Rice Biology, China National Rice Research Institute, Hangzhou 310006, China; huangderun@caas.cn (D.-R.H.); xiaojunwords@126.com (X.-J.N.); fanyeyang@caas.cn (Y.-Y.F.); zhangzhenhua@caas.cn (Z.-H.Z.); 3College of Life and Environmental Sciences, Hangzhou Normal University, Hangzhou 310012, China; yishen1912@163.com

**Keywords:** rice, grain shape, grain weight, RING-type E3 ubiquitin

## Abstract

Grain weight and grain shape are important traits that determine rice grain yield and quality. Mining more quantitative trait loci (QTLs) that control grain weight and shape will help to further improve the molecular regulatory network of rice grain development and provide gene resources for high-yield and high-quality rice varieties. In the present study, a QTL for grain length (GL) and grain width (GW), *qGL5.2*, was firstly fine-mapped into a 21.4 kb region using two sets of near-isogenic lines (NILs) derived from the *indica* rice cross Teqing (TQ) and IRBB52. In the NIL populations, the GL and ratio of grain length to grain width (RLW) of the IRBB52 homozygous lines increased by 0.16–0.20% and 0.27–0.39% compared with the TQ homozygous lines, but GW decreased by 0.19–0.75%. Then, by analyzing the grain weight and grain shape of the knock-out mutant, it was determined that the annotation gene *Os05g0551000* encoded a RING-type E3 ubiquitin ligase, which was the cause gene of *qGL5.2*. The results show that GL and RLW increased by 2.44–5.48% and 4.19–10.70%, but GW decreased by 1.69–4.70% compared with the recipient. Based on the parental sequence analysis and haplotype analysis, one InDel variation located at −1489 in the promoter region was likely to be the functional site of *qGL5.2*. In addition, we also found that the Hap 5 (IRBB52-type) increased significantly in grain length and grain weight compared with other haplotypes, indicating that the Hap 5 can potentially be used in rice breeding to improve grain yield and quality.

## 1. Introduction

Grain weight and grain shape are important traits that determine rice grain yield and quality. Mining more quantitative trait loci (QTLs) that control grain weight and shape will help to further improve the molecular regulatory network of rice grain development and provide gene resources for high-yield and high-quality rice varieties.

Up to now, the Gramene database (https://www.gramene.org/qtl, accessed on 11 June 2024) contains more than 400 QTLs for grain weight and shape in rice, of which 26 QTLs have been cloned. Among these cloned genes controlling grain weight and shape, there are 17 genes whose function changes are caused by the sequence variation in the coding region, including *qTGW1.2b*, *qGS1-35.2*, *GW2*, *GL2*/*GS2*, *OsLG3b*/*LGY3*, *GS3.1*, *GS3*, *SG3*, *GL3.1*/*qGL3*, *GSA1*, *GL3.3*/*TGW3*, *TGW6*, *GW6a*, *GL6*, *GLW7*, *GL7*/*GW7*, and *qTGW10*-20.8/*GL10*/*qGL10* [1,2,3,4,5,6,7,8,9,10,11,12,13,14,15,16,17,18,19,20,21,22,23,24,25]. There are three genes whose function changes are caused by the sequence variation in the promoter region, including *GW6a*, *GL6*, and *GLW7*. The functional changes of five genes are caused by the sequence variation in both the coding region and promoter region, including *GSE5*/*GW5*, *qGL5*, *GW6*, *GW8*, and *GW10* [25,26,27,28,29,30,31]. It was found that these cloned QTLs play a role in several signal pathways regulating cell elongation and proliferation, including the ubiquitin–proteasome pathway, G-protein signaling, plant hormone signaling, and a range of transcriptional regulators.

Ubiquitination is a common post-translational modification of protein in eukaryotes, which is widely involved in regulating many life processes such as organism growth and development, transcription regulation, responses to biotic and abiotic stresses, signal transduction, and so on [32,33,34]. E3 ubiquitin ligase is the key factor in the ubiquitin–proteasome pathway, which is divided into three subfamilies, including the HECT type, RING type, and U-box type [35,36,37,38]. HECT-type E3 ubiquitin ligase directly participates in the process of substrate ubiquitination [39]. RING-type and U-box-type E3 ubiquitin ligases are not directly involved in the process of substrate ubiquitination [40,41,42,43,44]. Up to now, 476 RING type E3 ubiquitin ligases have been identified in rice [45]. *GW2* located in the short arm of chromosome 2 is a negative regulator of 1000-grain weight and grain width, encoding a ring-type E3 ubiquitin ligase [3]. Hao et al. [41] showed that *GW2* induced glutathione protein WG1 to degrade in 26S proteasome, which eliminated the inhibition of WG1 on the transcription activity of *OsbZIP47* and promoted the transcription of downstream genes, thus regulating the grain weight and shape.

In a previous study, a minor-effect QTL for grain length (GL) and grain width (GW), *qGL5.2*, located within the interval Fi27369−Fi27390 on the long arm of rice chromosome 5 [46]. This study verified the genetic effects of *qGL5.2* on grain shape using two NIL populations developed from the *indica* rice cross Teqing (TQ)/IRBB52. Finally, the *qGL5.2* was fine-mapped to a 21.4 kb region, which contained two annotation genes. Then, by analyzing the grain weight and grain shape of the knockout mutant, it was determined that the annotation gene *Os05g0551000* encoded a RING-type E3 ubiquitin ligase, which was the cause gene of *qGL5.2*. In addition, based on the results of parental sequence analysis and haplotype analysis, one InDel variation located in the promoter region was likely to be the functional site of *qGL5.2*. Our findings provide a new gene resource to improve grain appearance quality.

## 2. Results

### 2.1. Fine-Mapping of qGL5.2

Firstly, two F_17:18_ NIL populations carrying the heterozygous region Fi27369−Fi27390, FW1, and FW2, were used to narrow down the region of the *qGL5.2*. As illustrated in Figure 1, FW1 and FW2 were derived from an F_15_ single plant of the cross TQ/IRBB52. Each of them consisted of 35 NIL-TQ homozygous lines and 35 NIL-IRBB52 homozygous lines in the segregating region. At maturity, four traits—1000-grain weight (TGW), grain length (GL), grain width (GW), and the ratio of grain length to grain width (RLW)—were measured.

The phenotypic distribution of the four traits was plotted as two series. For GL and RLW, the difference between the NIL-TQ and NIL-IRBB52 lines was obvious: the former was concentrated in the low-value area, while the latter was concentrated in the high-value area. For GW, the NIL-TQ lines were concentrated in the high-value area, and the NIL-IRBB52 lines were concentrated in the low-value area. For TGW, there was no obvious difference between the NIL-TQ and NIL-IRBB52 lines (Figure 2). These results suggested that the QTLs controlling GL and GW segregated in these two populations, with the TQ allele increasing GW, and the IRBB52 allele increasing GL.

Two-way analysis of variance (ANOVA) was performed to test the phenotypic differences in these two NIL populations. As shown in Table 1, in FW1 and FW2 populations, there were significant differences in GL, GW, and RLW between NIL-TQ and NIL-IRBB52 homozygous lines (*p* < 0.05). For GL and RLW, the increasing allele was derived from IRBB52, while the increasing allele of GW was derived from TQ. In these two populations, the highest contribution rate of *qGL5.2* to trait variation was RLW, followed by GL and GW, which were 13.12%, 7.43%, and 3.63% in the FW1 population and 18.85%, 10.67%, and 6.14% in FW2 population, respectively. The IRBB52 allele increased GL by 0.013 and 0.016 mm but decreased GW by 0.003 and 0.006 mm in FW1 and FW2, respectively. Since the IRBB52 allele increased GL and decreased GW, there was no significant difference in TGW between the NIL-TQ and NIL-IRBB52 homozygous lines. These results indicated that *qGL5.2* controlled both GL and GW, and the *R*^2^ to GL was greater than that to GW. Since the increasing alleles controlling GL and GW came from different parents, there was no significant difference for TGW in FW1 and FW2. Finally, *qGL5.2* was delimited into a 24.1 kb region flanked by Fi27369 and Fi27390 (Appendix A).

### 2.2. Genetic Effects of qGL5.2 in Knock-Out Mutants

Based on the RGAP database (https://rice.plantbiology.msu.edu, accessed on 3 July, 2024), there are two annotation genes in the 21.4 kb region, *Os05g0551000* and *Os05g05511000*. *Os05g0551000* encodes a RING-type E3 ubiquitin ligase *CLG1* [47], and *Os05g05511000* encodes an expression protein with an unknown function. Since a previous study showed that the *CLG1* targeted *GS3* for degradation via the endosome pathway to determine grain size in rice, we supposed that the *Os05g0551000* was the candidate gene for *qGL5.2*.

*Os05g0551000* knock-out (KO) mutants were produced to validate the effects of *qGL5.2*. Using the CRISPR/Cas9 system, four independent T_0_ homozygous mutants, KO1, KO2, KO3, and KO4, were produced from the recipient R780, a NIL carrying the IRBB52 allele in the *qGL5.2* segregating region. KO1, KO2, and KO3 had a 1 bp insertion (A) at +38 in the 1st exon, causing frameshift mutation at the 13th amino acid (AA) and premature termination at the 60th AA. KO4 had an 8 bp detection at the +35 position, which caused frameshift mutation at the 12th AA and premature termination at the 51st AA (Figure 3A). 

Four T_1_ populations and transgenic-negative (CK) were planted in HZ, with each population consisting of 24 plants. At maturity, four traits—TGW, GL, GW, and RLW—were measured. The results of the phenotypic differences between CK and mutants are shown in Table 2. Compared with CK, the GL and RLW of the four mutants increased significantly, but the GW decreased significantly (Figure 3B,C). Among the four mutants, the values ranged from 0.190 to 0.427 mm for GL, 0.117 to 0.297 for RLW, and −0.047 to −0.131 mm for GW, respectively. For TGW, KO1, KO3, and KO4 decreased significantly compared with CK, while KO2 had no significant difference compared with CK. These results imply that *Os05g0551000* is the causal gene for *qGL5.2*, hereafter referred to as *GL5.2*.

### 2.3. Genomic Sequence Analysis of GL5.2 between the Two Parents

Comparing the promoter and coding region sequences of *GL5.2* between the parents TQ and IRBB52, two SNPs in the coding region and one InDel in the promoter were detected. Using Nipponbare as a reference sequence, G696A in the third exon and C2265A in the ninth exon were found, but they did not cause amino acid variation. The InDel was found at −1489 in the promoter region, and the TQ allele had a 1 bp (T) insertion. Then, *cis*-acting regulatory elements in the promoter region of the two parents were predicted using the Plant CARE database. At the position of the InDel in the promoter region, no *cis*-acting element was predicted in the *GL5.2*^TQ^ allele, but one GT1-motif (GTGTGTGAA) element, found in *Solanum tuberosum*, was predicted in the *GL5.2*^IRBB52^ promoter.

### 2.4. Haplotype Analysis of GL5.2 Using the Natural Variations in the Promoter

The promoter regions of *GL5.2* in 3898 rice accessions were subjected to haplotype analysis using the resequencing data in the RiceVarMap V2.0 database. Ten haplotypes were classified and sorted according to the number of accessions (Appendix A). Based on the sequences of the two parents, TQ has Hap4, and IRBB52 has Hap5. In Hap1, Hap6, Hap7, and Hap8, the proportion of *indica* accessions was high, accounting for 97.9%, 89.9%, 95.4%, and 100.0%, respectively. In Hap2, the proportion of *japonica* accessions was high, accounting for 92.5%. In Hap4 (TQ type), the number of *AUS* accessions was higher than that of *indica* and *japonica* accessions. In Hap5 (IRBB52 type), the number of *japonica* accessions was more than that of *indica* accessions.

Subsequently, the phenotypic data of TGW, GL, GW, and RLW documented in RiceVarMap V2.0 was downloaded. The analysis for phenotypic differences among six haplotypes with more than five phenotypic data—1, 2, 4, 6, 7, and 9—was performed. For TGW and GL, Hap5 (IRBB52 type) had the highest trait value. For GW and RLW, Hap5 had the second highest. It showed that the Hap5 was a superior allele at this locus. For Hap4 (TQ type) and Hap5 (IRBB52 type), significant differences were detected on all four traits except RLW. Compared with Hap4 (TQ type), Hap5 (IRBB52 type) increased significantly in TGW, GL, and GW (Figure 4A–D). In addition, since only one InDel was detected between TQ and IRBB52 in the promoter of *GL5.2*, we divided the accessions into two groups according to this variation. Phenotypic differences were analyzed by using Student’s *t*-test. The TGW, GL, and RLW of the IRBB52 group were significantly higher than those of the TQ group (Figure 4E–H). These results suggested that the InDel in the promoter was likely to be the functional site of *GL5.2* to control grain size.

## 3. Discussion

In the present study, *qGL5.2* controlling grain length was fine-mapped into a 24.1 kb region on the long arm of chromosome 5. Then, it was determined that *Os05g0551000* encoded a RING-type E3 ubiquitin ligase, which was the cause gene of *qGL5.2*. All four knock-out mutants increased GL and RLW and decreased GW. Cloning of *GL5.2* provides a new gene resource for improving grain appearance quality in rice.

Grain weight and shape are the key traits that determine the yield and quality of rice. Grain weight is mainly determined by the grain size, which depends on the grain length, grain width, and grain thickness, and the grain length and grain width determine the grain shape. These traits are complex quantitative traits that are controlled by a small number of major-effect QTLs and a large number of minor-effect QTLs. Since the effect of minor QTL is small, phenotypic identification is easily affected by measurement errors. At present, only a few minor-effect QTLs controlling grain weight and shape, such as *qTGW1.2b*, *qTGW10-20.8*, and *qGS1-35.2*, have been cloned [1,2,23]. In this study, the *qGL5.2* was fine-mapped using two NIL populations derived from a residual heterozygote (RH) (Figure 1). A RH indicates that a single plant was segregating for the target region, but homozygous in other regions. Then, the candidate gene was knocked out by the CRISPR/Cas9 system. In the NILs, the GL of NIL-IRBB52 lines increased by 0.16–0.20% compared to NIL-TQ lines. In KO mutants, the GL of the mutants increased by 2.44–5.48% compared to CK. These results suggested that editing minor allelic variations can create new major allelic variations. This approach enhances the application value of minor-effect genes in rice breeding.

The results of the sequence comparison of the coding region showed that although there were two SNP variations between the two parents, they did not cause changes in amino acid variation. Nevertheless, sequence comparison of the promoter region showed that the TQ allele had a 1 bp (T) insertion at −1489. According to this InDel, 468 rice germplasms were divided into two groups. TGW, GL, and RLW of IRBB52-type were significantly higher than that of TQ-type (Figure 4E–H). In addition, by using the Plant CARE database, a GT1-motif found in *Solanum tuberosum* was predicted to be at the position of the variation in the IRBB52 promoter, but the TQ allele failed to predict this motif at this site due to the 1 bp insertion. The GT element was originally identified in the promoter of the *rbcS*-*3A* gene in pea leaves, and it had been found in many promoters encoding different functional genes [48,49,50]. The GT1-motif in *Solanum tuberosum is* related to photoreaction [51]. These results suggested that the InDel in the promoter region may be the functional site of *GL5.2*. Further study is needed on how this element regulates grain shape.

Gene function may change in different genetic backgrounds. The non-functional allele of *GL3.3* increased grain length by 2.00 mm in a *japonica* background, but only increased grain length by 0.49 mm in an *Aus* background. Overexpression of the grain weight and size gene *OsMADS56* in a *japonica* rice variety HJX did not change the grain width, but in an *indica* rice variety ZY180 it increased the grain width. Furthermore, the grain weight and size gene *OsPUB3* was knocked out in different backgrounds, and the grain length of the KO mutant increased and decreased, respectively. For *GL5.2*, in a previous study, Yang et al. [47] knocked out *CLG1* in Zhonghua 11 (a *japonica* rice variety), and the grain length of the mutants did not change significantly. In this study, we knocked out the *GL5.2* in an *indica* rice background, and the grain length of the mutants increased significantly. These results suggested that the genetic effect of *CLG1*/*GL5.2* was influenced by the genetic background.

The results of haplotype analysis showed that the number of the *indica* and *japonica* rice accessions was the highest in Hap1 and Hap2, respectively. The RLW of Hap1 was significantly higher than that of Hap2 (Figure 4D). These results indicated that Hap1 and Hap2 were the superior haplotypes in *indica* and *japonica* rice, respectively, after long-term artificial or natural selection. In addition, compared with Hap1 and Hap2, Hap5 had significantly increased GL and TGW. In the improvement of rice breeding, GL and TGW are closely related to the rice appearance quality and yield, respectively. These results implied that Hap5 could be used not only for the rice appearance quality improvement of *japonica* rice, but also for the yield improvement of *indica* rice, and had an important application potential.

## 4. Materials and Methods

### 4.1. Development of NIL Populations

Two NIL populations segregating in an isogenic background were used to narrow down the region of *qGL5.2*. As shown in Figure 1, one F_15_ single plant from the cross TQ/IRBB52 with the heterozygous region Fi27369–Fi27390 covering the *qGL5.2* was first selected. In the resultant F_16_ population, two plants, heterozygous in Fi27369 and Fi27390, respectively, were identified and selfed to produce two F_17_ populations. In each population, homozygous non-recombinants were identified and selfed to develop homozygous lines. Two F_17:18_ NIL populations named FW1 and FW2 were constructed and used for fine-mapping of *qGL5.2*.

### 4.2. Generation of Knock-Out Mutants

The CRISPR/Cas9 system was performed to construct knock-out mutants for the annotated gene, *Os05g0551000*. One target, located at +21 to +43 in the 1st exon (Figure 3), was selected using the CRISPR-GE system (https://skl.scau.edu.cn, accessed on 13 October 2021). Oligonucleotide cri-1 (Appendix A) was designed and ligated into BGK03 vector according to the manufacturer’s instruction (BIOGLE Co., Ltd., Hangzhou, China). The original BGK03 vector comprised a rice U6 promoter for activating the target site sequence, a Spcas9 gene driven by the maize ubiquitin promoter, and a hygromycin marker gene driven by Cauliflower mosaic virus 35S promoter. Then, the CRISPR/Cas9 constructs were introduced into recipient R780 using *Agrobacterium tumefaciens*-mediated transformation through rice calli, which was performed by BioRun Co., Ltd. (Wuhan, China). Twelve independent T_0_ plants were obtained. The genomic DNA of the T_0_ plants was extracted from young leaves using the DN easy Plant Mini Kit. Ten Hyg-positive transgenic plants were detected by using the hygromycin gene marker Hyg (Appendix A). The genomic fragment containing the target region was amplified using the sequencing markers seqcri-1 (Appendix A). The PCR products were sequenced by the Sanger method and decoded using the web-based tool DSDecodeM (https://skl.scau.edu.cn/dsdecode, accessed on 10 May 2022). Among the ten independent T_0_ plants tested, four of them were homozygous mutants, five were heterozygous mutants and one showed no mutation. Finally, four homozygous T_0_ mutants and one transgenic-negative were obtained. Each T_0_ plant was selfed to produce a T_1_ population.

### 4.3. Field Experiments and Phenotyping

All the rice materials were planted in the paddy field of the China National Rice Research Institute in Hangzhou, Zhejiang province, China. 

The two NIL populations were grown following a randomized complete block design with two replications. In each replication, each line was planted in a single row of 10 plants, with 26.7 cm between rows and 16.7 cm between plants. Field management followed normal agricultural practices. At maturity, four of the middle eight plants in each row were harvested. Fully filled grains were selected and measured for TGW, GW, GL, and RLW following the method reported by Zhang et al. [52]. In brief, the grains were soaked in 3.5 mol/L NaCl solution and the floating grains were removed. Around 18 g of fully filled grains were obtained and then dried. These grains were split into two halves and TGW, GL, and GW were measured using an automatic seed counting and analyzing instrument (Model SC-G, produced by Wanshen Ltd., located in Hangzhou, China).

The four T_1_ populations of the *GL5.2* KO mutants and the CK were grown with no replication and measured on a single-plant basis. Each of the four populations contained 24 individual plants.

### 4.4. Data Analysis

Two-way ANOVA was used to test phenotypic differences in the NIL populations. The analysis was performed using the statistical analysis software SAS (V8) [53]. A mixed model GENOTYPE + LINE (GENOTYPE) + REP + GENO-TYPE*REP was applied, in which LINE (GENOTYPE) was defined as a random effect and used as the error term to test GENOTYPE differences. When significant differences were detected (*p* < 0.05), the additive effect was estimated by (IRBB52–TQ)/2. Positive values indicate that the increasing allele is from IRBB52, and negative values indicate that the increasing allele is from TQ. Student’s *t*-test was employed to test the phenotypic differences (*p* < 0.05) between the KO mutants and CK. Duncan’s multiple range test was used to determine the phenotypic differences among the haplotypes (*p* < 0.05).

## 5. Conclusions

One QTL for grain length, *qGL5.2*, was fine-mapped within the 21.4 kb region on chromosome 5 and its casual gene, *GL5.2*, was validated by using CRISPR/Cas9. Knockout of *GL5.2* exhibits positively regulated grain weight by controlling grain length in rice. One InDel difference was found in the promoter region between the two parents. The haplotype of the male parent increased significantly in grain length and grain weight compared with other haplotypes owing to the haplotype analysis, indicating that *GL5.2* can potentially be used in rice breeding to improve grain yield and quality.

## Figures and Tables

**Figure 1 plants-13-02521-f001:**
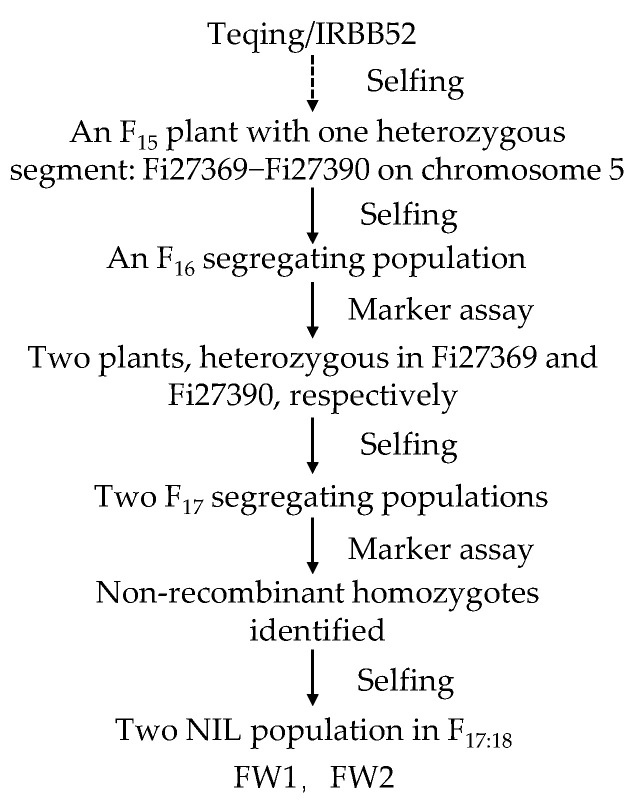
Development of the rice populations used in this study. NIL, near-isogenic line.

**Figure 2 plants-13-02521-f002:**
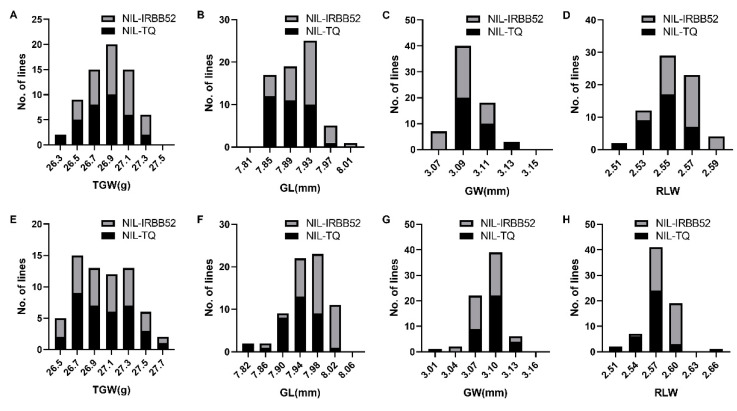
Distributions of 1000 grain weight, grain length, grain width, and ratio of grain length to grain width in two near-isogenic line populations. (**A**–**D**) FW1; (**E**–**H**) FW2. TGW, 1000 grain weight; GL, grain length; GW, grain width; RLW, ratio of grain length to grain width; NIL-TQ and NIL-IRBB52 are near-isogenic lines having Teqing and IRBB52 homozygous genotypes in segregating region, respectively.

**Figure 3 plants-13-02521-f003:**
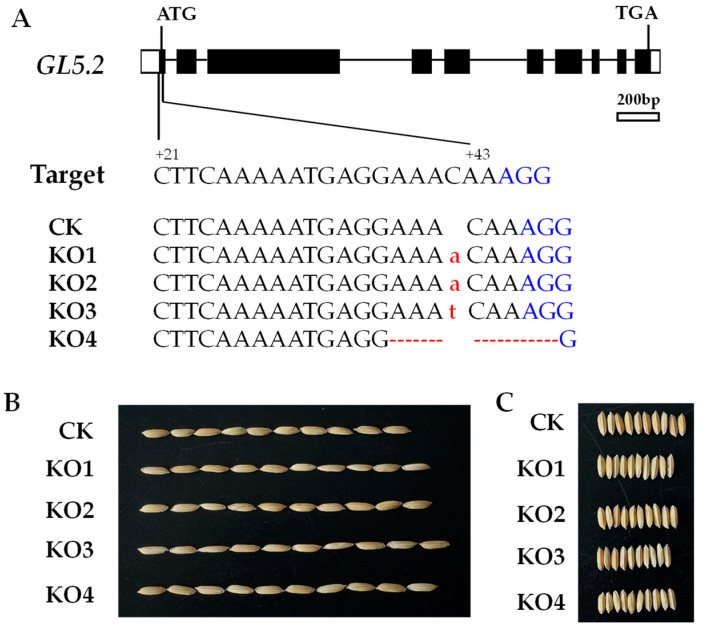
Sequence and phenotypic variation between the knock-out mutants and the transgenic-negative. (**A**) Variations of the DNA sequences in the target region. The protospacer adjacent motif site is shown in blue. Insertion is indicated by lowercase letter in red. Deletion is indicated by a hyphen in red; (**B**) Grain length of the CK and knock-out mutants; (**C**) Grain width of the CK and knock-out mutants. CK, transgenic-negative.

**Figure 4 plants-13-02521-f004:**
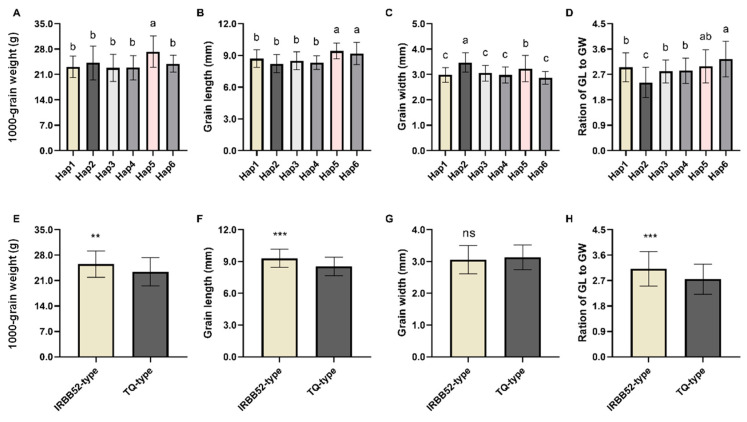
Phenotypic differences among different groups of *GL5.2*. (**A**–**D**), Phenotypic differences among six haplotypes of *GL5.2*. Values are given as the mean ± SD (n = 135 for Hap1, n = 116 for Hap2, n = 103 for Hap3, n = 36 for Hap4, n = 12 for Hap5, n = 11 for Hap6). Values with different letters are significantly different at *p* < 0.05 based on Duncan’s multiple range test; (**E**–**H**), Phenotypic differences between IRBB52-type and TQ-type of *GL5.2*. Values are given as the mean ± SD (n = 441 for IRBB52-type; n = 27 for Teqing-type). **, *p* < 0.01; ***, *p* < 0.001; ns: not significant.

**Table 1 plants-13-02521-t001:** Fine-mapping of *qGL5.2* using two NIL populations.

Population	Trait	Phenotype (Mean ± SD)		*p*	*A*	*R*^2^ (%)
		NIL-TQ	NIL-IRBB52			
FW1	TGW	26.82 ± 0.26	26.89 ± 0.28	0.9132		
	GL	7.890 ± 0.034	7.915 ± 0.038	0.0134	0.013	7.43
	GW	3.098 ± 0.014	3.090 ± 0.012	0.0401	−0.003	3.63
	RLW	2.547 ± 0.016	2.561 ± 0.015	0.0003	0.007	13.12
FW2	TGW	26.99 ± 0.31	26.91 ± 0.41	0.3669		
	GL	7.938 ± 0.044	7.972 ± 0.039	0.0028	0.016	10.67
	GW	3.096 ± 0.017	3.084 ± 0.023	0.0128	−0.006	6.14
	RLW	2.565 ± 0.018	2.585 ± 0.019	<0.0001	0.010	18.85

TGW, 1000-grain weight (g); GL, grain length (mm); GW, grain width (mm); RLW, ratio of length to grain width; *A*, additive effect of replacing a Teqing allele with an IRBB52 allele; *R*^2^, proportion of the phenotypic variance explained by the QTL; NIL-TQ and NIL-IRBB52, near-isogenic lines with Teqing and IRBB52 homozygous genotypes in the segregating region, respectively.

**Table 2 plants-13-02521-t002:** Genetic effects of *qGL5.2* in four knock-out mutants.

Trait	Population	Phenotype (Mean ± SD)	±CK	±CK (%)
TGW	CK	22.61 ± 0.42		
	KO1	22.28 ± 0.61	−0.34 *	−1.49
	KO2	22.53 ± 0.54	−0.08	−0.34
	KO3	22.10 ± 0.56	−0.51 ***	−2.27
	KO4	22.08 ± 0.56	−0.53 ***	−2.35
GL	CK	7.788 ± 0.105		
	KO1	7.995 ± 0.089	0.207 ****	2.65
	KO2	7.978 ± 0.079	0.190 ****	2.44
	KO3	8.215 ± 0.104	0.427 ****	5.48
	KO4	8.017 ± 0.095	0.229 ****	2.93
GW	CK	2.803 ± 0.030		
	KO1	2.727 ± 0.034	−0.076 ****	−2.72
	KO2	2.756 ± 0.031	−0.047 ****	−1.69
	KO3	2.672 ± 0.043	−0.131 ****	−4.70
	KO4	2.736 ± 0.035	−0.067 ****	−2.42
RLW	CK	2.779 ± 0.059		
	KO1	2.932 ± 0.048	0.153 ****	5.52
	KO2	2.895 ± 0.033	0.117 ****	4.19
	KO3	3.076 ± 0.074	0.297 ****	10.70
	KO4	2.931 ± 0.031	0.152 ****	5.47

TGW: 1000-grain weight (g); GL: grain length (mm); GW: grain width (mm); RLW: ratio of grain length to grain width. ±CK: Values increase or decrease over the transgenic negative control CK. ±CK (%): Percentage increase or decrease over the transgenic negative control CK. Values are given as the Mean ± SD. Phenotypic differences between mutants and CK were tested using Student’s *t*-test. *, *p* < 0.05; ***, *p* < 0.001; ****, *p* < 0.0001.

## Data Availability

The original contributions presented in the study are included in the article/Appendix A, further inquiries can be directed to the corresponding author.

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
