# Peer review of "GL5.2, a Quantitative Trait Locus for Rice Grain Shape, Encodes a RING-Type E3 Ubiquitin Ligase"

_plants, 2024, doi:10.3390/plants13172521_

Round 1
Reviewer 1 Report
Comments and Suggestions for Authors
The manuscript by Zhang et al. entitled “GL5.2, a QTL for rice grain shape encodes a RING-type E3 ubiquitin ligase” report the first fine-mapped region that annotates the gene Os05g0551000 encoded a RING-type E3 ubiquitin ligase related grain length and width. The authors phenotyped the grain traits by comparison of the KO mutants and Wild-type control. The work is technically sound piece of research, and within the scope of Plant, however, some issues need to be fixed before acceptance.
1. Which cas9 system in this research? Spcas9, Sacas9 or others
2. Seed phenotyping is a systemic approach that need target seeds harvesting from various conditions, for example, different nitrate/nitrite application in the test fields that should be in different climate area. In this research, the authors simply mentioned the rice cultivation in ‘Field Experiments and Phenotyping’ and seed phenotype evaluation, that may be led a misunderstanding on the results of the seed phenotyping process.
Comments on the Quality of English Languageaverage
Author Response
Comments 1:Which cas9 system in this research? Spcas9, Sacas9 or others
Response 1: Thank you. Spcas9 was used in this research. We have made revisions (Lines 273).
Comments 2:Seed phenotyping is a systemic approach that need target seeds harvesting from various conditions, for example, different nitrate/nitrite application in the test fields that should be in different climate area. In this research, the authors simply mentioned the rice cultivation in ‘Field Experiments and Phenotyping’ and seed phenotype evaluation, that may be led a misunderstanding on the results of the seed phenotyping process.
Response 2: Thank you. We are apologized for the unclear description of seed phenotyping process. Information on the seed phenotyping process has been added (Lines 295−299).
Reviewer 2 Report
Comments and Suggestions for Authors
The manuscript under consideration is devoted to the analysis of the qGL5.2 locus associated with some effects of grain length and width control in rice and is a continuation of the work previously carried out by the team with this locus. Studies on the identification and analysis of QTL loci, which include several genes affecting the expression of quantitative traits, is relevant and significant for improving economically valuable characteristics in agricultural crops. In the presented manuscript, the authors focus on clarifying the localization and functional significance of this locus. For this purpose, two mapping populations of rice were created, with the help of which the authors were able to successfully localize the region of interest, determine its size (21.4 kb) and, using bioinformatics approaches, find two protein-coding sequences in this region. The function of one of these sequences was unknown, while the second corresponded to the Os05g551000 gene encoding RING type 2 E3 ubiquitin ligase. The authors then attempted to answer the question of whether this gene makes any contribution to the change in rice grain shape parameters. For this purpose, four knockouts were obtained using the CRISPR/Cas9 approach and, based on the analysis results, it was concluded that the refined GL5.2 locus is a positive regulator of such parameters as GL and RLW. Based on the haplotype analysis results, Hap5 was identified, which is associated with an increase in the length and weight of rice grains, which can be used in the breeding of this important agricultural crop. The submitted manuscript can be recommended for publication in the journal, taking into account the comments below.
Comments:
1. The authors indicated in section 4.2. (lines 277-278) that the created tool for genomic editing was delivered to the rice plant genome using Agrobacterium-mediated transformation. Since the authors did not provide a reference to the Agrobacterium-mediated transformation method, it remains unclear whether rice calli or the floral dip method were used for transformation. It is necessary to provide a reference to the work describing the rice transformation technique.
2. Section 4.2. (line 278) states that a total of 12 To plants were obtained after agrobacterial transformation, among which Hyg-positive ones were selected and among the latter, four To homozygous mutants were selected (line 283). Since the PCR analysis results are not provided, it is completely unclear how homozygotes were obtained after To editing? During genome editing, nuclease usually makes a cut in the region of one allele of the gene being studied. Although two cuts along two alleles are possible, this is not a frequent event. In this work, it remains unclear how To homozygotes were obtained. It is necessary to provide evidence of the homozygous nature of the To plants used further to obtain T1 and conduct haplotype analysis.
Author Response
Comments 1: The authors indicated in section 4.2. (lines 277-278) that the created tool for genomic editing was delivered to the rice plant genome using Agrobacterium-mediated transformation. Since the authors did not provide a reference to the Agrobacterium-mediated transformation method, it remains unclear whether rice calli or the floral dip method were used for transformation. It is necessary to provide a reference to the work describing the rice transformation technique.
Response 1: Thank you. We are apologized for the unclear description of the Agrobacterium-mediated transformation method. The rice calli was used for transformation. We have made revisions (Lines 276−277).
Comments 2: Section 4.2. (line 278) states that a total of 12 plants were obtained after agrobacterial transformation, among which Hyg-positive ones were selected and among the latter, four homozygous mutants were selected (line 283). Since the PCR analysis results are not provided, it is completely unclear how homozygotes were obtained after editing? During genome editing, nuclease usually makes a cut in the region of one allele of the gene being studied. Although two cuts along two alleles are possible, this is not a frequent event. In this work, it remains unclear how homozygotes were obtained. It is necessary to provide evidence of the homozygous nature of the plants used further to obtain T1 and conduct haplotype analysis.
Response 2: Thank you for your comment. In our study, the genomic fragment containing the target region was amplified. And then, the PCR products were sequenced by the Sanger method and decoded using the web-based tool DSDecodeM (https://skl.scau.edu.cn/dsdec ode). Among the ten independent T0 plants tested, four of them were homozygous mutants, five were heterozygous mutants and one showed no mutation. (Lines 282−285).

Round 2
Reviewer 2 Report
Comments and Suggestions for Authors
The authors of the manuscript under review have taken into account the reviewer's comments and made appropriate changes to the text of the manuscript. This manuscript can be recommended for publication in the journal.